# Se-Enrichment Pattern, Composition, and Aroma Profile of Ripe Tomatoes after Sodium Selenate Foliar Spraying Performed at Different Plant Developmental Stages

**DOI:** 10.3390/plants10061050

**Published:** 2021-05-23

**Authors:** Annalisa Meucci, Anton Shiriaev, Irene Rosellini, Fernando Malorgio, Beatrice Pezzarossa

**Affiliations:** 1Institute of Life Sciences, Sant’Anna School of Advanced Studies, 56127 Pisa, Italy; annalisa.meucci@santannapisa.it; 2Research Institute on Terrestrial Ecosystems, 56124 Pisa, Italy; irene.rosellini@cnr.it (I.R.); beatrice.pezzarossa@cnr.it (B.P.); 3Department of Agriculture, Food and Environment, University of Pisa, 56124 Pisa, Italy; fernando.malorgio@unipi.it

**Keywords:** biofortification, fruit quality, ripening, selenium, *Solanum lycopersicum*, volatile organic compounds (VOCs)

## Abstract

Foliar spray with selenium salts can be used to fortify tomatoes, but the results vary in relation to the Se concentration and the plant developmental stage. The effects of foliar spraying with sodium selenate at concentrations of 0, 1, and 1.5 mg Se L^−1^ at flowering and fruit immature green stage on Se accumulation and quality traits of tomatoes at ripening were investigated. Selenium accumulated up to 0.95 µg 100 g FW^−1^, with no significant difference between the two concentrations used in fruit of the first truss. The treatment performed at the flowering stage resulted in a higher selenium concentration compared to the immature green treatment in the fruit of the second truss. Cu, Zn, K, and Ca content was slightly modified by Se application, with no decrease in fruit quality. When applied at the immature green stage, Se reduced the incidence of blossom-end rot. A group of volatile organic compounds (2-phenylethyl alcohol, guaiacol, (E)-2-heptenal, 1-penten-3-one and (E)-2-pentenal), positively correlated with consumer liking and flavor intensity, increased following Se treatment. These findings indicate that foliar spraying, particularly if performed at flowering stage, is an efficient method to enrich tomatoes with Se, also resulting in positive changes in fruit aroma profile.

## 1. Introduction

Selenium plays several crucial roles in human metabolism [1,2,3]. At the right concentrations, it positively affects DNA synthesis, fertility, reproduction, and muscle function. Selenium helps to slow down aging, prevent certain cancers, and reduce the incidence of viral infections, cardiovascular damage, arthritis, and alterations of the immune system [4]. Due to its ability to oxidize thiol groups in the virus protein disulfide isomerase, selenite may even prevent COVID-19 contagion [5].

The recommended dietary allowance (RDA) for selenium is 55 µg day^−1^, and the tolerable upper intake for adults is 400 µg day^−1^ [6]. The European Food Safety Authority set the RDA for adults at 70 µg Se day^−1^ [7].

Worldwide, approximately one in eight people may have insufficient Se intake [8]. Those living in areas with low selenium in the soil, and consequently low levels in their daily diet, need to replenish a deficiency through medicines or nutraceuticals. Agronomic biofortification with Se fertilization is considered an effective way to produce Se-rich crops, thus improving the Se intake in the target population [9,10,11].

Selenium is not considered an essential element in plants, but it elicits the production of secondary metabolites and increases the enzymatic and non-enzymatic antioxidant capacity. Thus, in addition to biofortification, Se enhances the levels of specific bio-active/health-promoting compounds that may also have positive effects on the plant physiology and metabolism [12,13]. In fact, plants supplemented with Se can better counteract oxidative stress as well as abiotic (salt, drought) and biotic stresses [14].

Although the exact mechanisms of Se action are still poorly understood [15], positive effects of Se on delaying senescence and, in crops bearing fleshy fruits, modulating ripening and metabolic profiling and composition of the produce have been reported [16,17,18,19,20,21,22,23].

Se-enrichment of fruits to be consumed as fresh produce is of great interest since some selenium is lost after cooking or processing [24]. Given that tomatoes (*Solanum lycopersicum*) are mostly consumed fresh and do not require heat processing, and since tomato is the second-most important vegetable crop worldwide, it is one of the best commodities for producing a selenium biofortified fruit for dietary supplementation.

Various strategies can be used for the Se-biofortification of different crops, including tomato [14]. Se supplementation (as sodium selenate or sodium selenite) to tomato plants via the root system (hydroponic, soils, or artificial substrate fertilization) enriches tomato fruit [13,18,20,25,26].

However, with the global focus on sustainability, ensuring that such strategies are safe for the environment is challenging. Excessive remaining chemicals in the growth substrate or solution, which are not taken up by plants, may cause water pollution [27] or increase soil salinity [28].

Foliar spraying is a feasible alternative, but few papers have investigated the effects of this method on Se-enrichment and on the metabolic, compositional, and physiological changes occurring in tomato fruit [13,19,22,29,30]. The results of these studies demonstrate that the enrichment effects of foliar treatments on tomato plants depend not only on the Se concentration and chemical form but also on the plant developmental stage when treatment is performed.

Since the range of Se concentration between essentiality and toxicity for humans and for plant tissues is quite narrow [31], the protocols need to be designed to enrich tomatoes with no risks for the consumers and no negative effects on the plants.

We investigated the responses to sodium selenate foliar spraying at flowering (FL) and fruit immature green stage (IG) in terms of Se accumulation in tomato fruit, and specific composition and quality-related traits at ripening, with a specific focus on volatile organic compound (VOC) profiles.

## 2. Results and Discussion

### 2.1. Selenium Concentration

Spraying tomato plants with sodium selenate resulted in a significant increase in selenium concentration in the fruit of both trusses (Table 1). The time of treatment did not affect the Se content in the fruit of the first truss, whereas in the second truss, the treatment performed at FL gave a higher selenium concentration compared to the IG treatment. The two doses of Se applied (1 and 1.5 mg L^−1^) did not result in statistically different accumulations of Se in the fruit tissues.

The selenium concentration in the fruits of sprayed plants was much lower than the fruits of plants grown in the nutrient solution supplied with sodium selenate at the same concentration [18,20]. In fact, in those two studies, the uptake of selenium by roots, and its translocation to the aerial part, led to a higher accumulation of Se in fruit.

In plants sprayed with selenate, the composition of the cuticle (epicuticular wax or the deposition of wax platelets) and the leaf tomentosity could have hampered selenium absorption. The lower Se absorption could also be ascribed to the absence of adhesive substances in the spraying solution, which generally facilitate better penetration of the active ingredients and prevent their washout.

The selenium concentration in the fruit of plants treated with 1 mg Se L^−1^ was lower than that reported by Zhu et al. [29], i.e., 500 μg Se kg^−1^, in plants treated with the same dose of selenium at the onset of flowering. Different growing and environmental conditions, as well as different genotypes, might explain the observed discrepancy of results.

The selenium content in 100 g of fresh tomatoes was calculated by multiplying the concentration of Se by the percentage of dry matter. Values ranged between 0.62 and 0.95 µg Se 100 g FW^−1^ in the fruit of the first truss and between 0.35 and 0.66 µg Se 100 g FW^−1^ in the fruit of the second truss (data not shown). Thus, the daily consumption of treated tomatoes appears to increase the selenium supplementation, with no risk of toxicity for humans.

### 2.2. Mineral Element Content

The enrichment with selenium affected the content of all the minerals, except for Fe in the fruit of the first truss, and for K in fruit of the second truss (Table 2 and Table 3).

In the first truss, the supply with the higher dose of selenium increased the content of Ca when plants were treated at IG stage, whereas it decreased the content of Cu, Ca, and Zn in the fruit of plants sprayed at FL and of K in plants sprayed at the IG stage.

Selenium treatment decreased the content of all the minerals, except for Fe, in fruit of the second truss. To the best of our knowledge, only one study has been conducted on the effects of foliar spraying with selenium on the mineral content of tomato fruit [26]. The results of that study showed that foliar spraying at 10 and 20 mg Se L^−1^, repeated every 20 days, did not modify the content of Ca, Mg, and K.

### 2.3. Effect of Selenium on Blossom-End Rot

A statistical analysis of data relating to blossom-end rot highlights that the biofortification with selenium at IG phase, at both concentrations, significantly reduced the incidence of this rot, but only in fruit of the second truss (Table 4).

Blossom-end rot is believed to be caused by a calcium imbalance within the plant, but in our trial, the selenium treatment did not increase the calcium content in fruit. Factors other than Se-concentration and/or Ca content should thus be considered in terms of the incidence of this physiological disorder.

### 2.4. Qualitative Characteristics of Fruit

#### 2.4.1. Fruit Composition

Biofortification with selenium, both at FL and IG, affected the compositional traits of fruit collected at the red-ripe stage. In the fruit of the first truss, Se enrichment decreased titratable acidity and increased dry weight compared to the control (Table 5), while SSC (soluble solid content) and maturity index were not affected. In the fruit of the second truss (Table 6), dry weight and titratable acidity showed the same effects as Se-enrichment, as observed in the first truss, whereas SSC increased. Thus, the maturity index was significantly affected by Se-enrichment.

The taste index, based on brix and titratable values, slightly decreased in fruit treated with 1.5 mg Se L^−1^ of both protocols and in fruit treated with 1 mg Se L^−1^ at the IG stage. Considering that a taste index of <0.7 is associated with low quality [32], plants treated with selenium selenate produced tasty fruit. These results on fruit composition agree with previous studies conducted by our research group [18,20] in which tomato plants grown in hydroponics were enriched with the same concentration of Se, as sodium selenate, which was added to the nutrient solution.

#### 2.4.2. Aroma Profiles

Fresh tomato fruit produces several hundred VOCs, some of which are major contributors to the overall aroma and consequently affect consumer experience [33]. Specific pathways are involved in the metabolism of the VOCs, and their concentration in ripe fruit depends on factors affecting fruit development and physiology. To assess whether Se-enrichment affects this important quality trait, we carried out specific VOC analyses on red-ripe fruit of the second truss from the control and from plants treated with 1.0 and 1.5 Se L^−1^ at FL. A total of 39 VOCs were identified in the three sets of samples (Appendix A).

Partial least squares regression analysis indicated that sodium selenate treatment affects the volatile profile of the tomatoes at ripening (Figure 1). Overall, the validated PLSDA (Partial least squares discriminant analysis) model explains about 90% (first and second factors together) of the variability recorded between sample groups. In fact, a clear segregation of the different treatments was observed, with Se-enriched samples clustering separately from the control tomatoes and from each other. This clustering indicates important differences in terms of the numbers of VOCs specifically induced by the different levels of Se applied at the flowering stage. In addition, the fact that the analysis was repeatable was highlighted by the correct positioning of the different analyzed fruit replicates, with samples from the same treatment being close together.

Lipoxygenase pathway-related compounds, such as hexanal, (Z)-3-hexenal and 1-hexanol, were associated with control fruit. Specific C5 and C7 compounds and derivatives instead appeared to be more abundantly accumulated in Se-biofortified fruit. In particular, (E)-2-heptenal and methyl heptenone, together with phenylethyl alcohol, characterized the 1.5 mg Se L^−1^-treated fruit. On the other hand, 1-penten-3-ol, 1-penten-3-one, and (E)-2-pentenal sit, in the PLSDA score plot, closer to the 1.0 mg Se L^−1^-treated samples, which also showed a strict association with guaiacol and methyl salicylate.

2-phenylethyl alcohol, guaiacol, and (E)-2-heptenal, together with 1-penten-3-one and (E)-2-pentenal, were identified by Tieman et al. [33] as compounds positively correlated with consumer liking and overall flavor intensity, with decreased accumulation in modern cultivars compared to heirloom *S. lycopersicum* varieties. Thus, Se-enriched tomatoes may possibly also benefit from the treatment in terms of organoleptic properties and more attractive traits for consumers. Of particular interest appears to be the increase in methyl salicylate characterizing the fruit treated with 1 mg Se L^−1^. This compound is one of several phenylpropanoids that significantly contribute to the unique flavor of tomato fruit and is also considered to be better than salicylic acid as the signal molecule transmitted in the systemic acquired resistance (SAR) response [34]. Together with linalool, methyl salicylate is involved in the immune response against *Pseudomonas syringae* bacterial attack [35].

The association of both compounds with selenium treatments rather than with control clusters may indicate a better immunity response of fruit treated with selenium against some bacterial diseases. The association of methylated compounds (methyl salicylate and methyl heptanone) with Se-treated fruit may show how selenium affects tomato methyltransferases. In fact, in a proteomics study, Zeng et al. [36] demonstrated that methyl transferases are among the differentially expressed proteins in naturally selenium-enriched rice.

## 3. Materials and Methods

### 3.1. Experimental Set-Up

The experiment was conducted in a greenhouse located at the Department of Agricultural, Food, and Agro-Environmental Sciences (DISAAAa) of the University of Pisa (PI), characterized by a controlled temperature system (13–30 °C) and a ventilation air temperature of 27 °C. The seedlings of *Solanum lycopersicum* var. Kreos were provided in rock wool cubes (GRODAN, ROCKWOOL Group, 75 × 75 × 65 mm) supplied with a nutrient solution for about a week before being transferred to rock wool slabs (1 × 0.15 × 0.75 m).

Plants were grown in a hydroponic system with a density of 3 plants m^−2^. They were trimmed above the second truss. Pollination was performed by mechanical vibration of the inflorescences. The nutrient solution, characterized by E.C. of 2.6 mS cm^−1^ and pH value of 5.6, was distributed through a drip irrigation system, three times a day in the early phenological phases of plant growth, and five times a day in the final phases. The composition of the nutrient solution was the following: 14 mM L^−1^ N-NO_3_, 1 mM L^−1^ N-NH_4_, 1 mM L^−1^ P, 8 mM L^−1^ K, 5 mM L^−1^ Ca, 1.6 mM L^−1^ Mg, 5.23 mM L^−1^ Na, 1.71 mM L^−1^ S-SO_4_, 3.43 mM L^−1^ Cl, 15 μM L^−1^ Fe, 20 μM L^−1^ B, 1 mM L^−1^ Cu, 5 μM L^−1^ Zn, 10 μM L^−1^ Mn, and 1 μM L^−1^ Mo.

The experimental scheme was organized in randomized blocks. Selenium was sprayed as sodium selenate solution (Sigma Aldrich, colorless crystalline powder) at concentrations of 0.1 and 1.5 mg Se L^−1^. Each plant was treated with 250 mL of Se-enriched solution, whereas each control plant was sprayed with 250 mL of distilled water. When the solution was supplied, the surrounding plants were shielded with plastic sheets to avoid contamination. For each selenium dosage, half of the plants were treated at flowering (FL) (June), and the other half at the immature green (IG) fruit stage (July). Fruits were harvested for all the analyses at the red ripening stage.

After harvesting, the selenium content in the fruit of both trusses was analyzed. Composition, taste index, micro- and macro element concentrations, and VOC profiles of fruit were determined.

### 3.2. Selenium and Mineral Content Analyses

Total selenium and micro- and macro- element concentrations were determined in the fruit of Se-enriched and control plants. Collected fruit samples were oven-dried at 50 °C and then ground in a mortar. A total of 0.5 g of powder for each replicate was digested with nitric and perchloric acids. The mineral elements (Cu, Zn, Mn, Fe, K, Ca, Mg) were determined with a fast sequential atomic absorption spectrometer (AA240FS, Agilent). To calculate the selenium content, the digests were reduced by hydrochloric acid, following Zasoski and Burau [37], and the atomic absorption spectrometer was coupled to a hydride generator (Varian VGA 77) [38].

### 3.3. Fruit Composition and Quality Parameters

Fresh and dry weights of harvested fruit were determined. Titratable acidity was measured on 10 mL of tomato juice with an automatic titrator, using sodium hydroxide (0.1 M NaOH) as the titling and phenophthalein as the indicator. The result was expressed in g of citric acid per 100 mL of juice. Soluble solid content, expressed in °Brix, was investigated using a digital refractometer (model 53011, Turoni, Forlì, Italy). Maturation and taste indexes were calculated using Navez et al.’s [32] formula, on the basis of °Brix, and titratable acidity as proposed by Hernandez Suarez et al. [39]:Maturity Index = ° Brix acidity(1)
Taste Index = (° Brix/20 × acidity) + acidity(2)

We also tried to detect the possible effect of Se dosage and time of treatment on the incidence of blossom-end rot. The percentage of fruit affected by blossom-end rot was evaluated by calculating the percentage of fruit that showed the symptoms on the total of fruit present at each flowering stage. Percentage data were then transformed into angular values before statistical analysis.

### 3.4. HS-SPME-GC-MS Analysis

For the VOC analysis, a slightly modified method optimized by Brizzolara et al. [40,41] was used. A total of 15 g of pericarp tissue (without seeds) were homogenized with 15 mL 1M NaCl water solution in a 50 mL plastic tube using T25 Ultra-Turrax (IKA, Königswinter, Germany) homogenizer. The obtained puree was frozen in liquid nitrogen and stored at −80 °C. Three replicates per treatment were analyzed for both the red ripe stage with a different level of Se accumulation in fruit.

VOCs were identified using a gas-chromatograph (Perkin Elmer Clarus 680) coupled with a mass spectrometer (Perkin Elmer Clarus 600 S). A total of 5 g of tomato puree was transferred into a 20 mL crimp vial (Sigma Aldrich, Milano, Italy) and was thawed at 15 °C for 15 min. Samples were incubated for 60 min at 40 °C, and VOCs were extracted for 45 min at the same temperature using an SPME Fiber (50/30 μm, DVB/CAR/PDMS, Sigma Aldrich, Italy).

We used the following GC temperature program: the initial temperature was 40 °C, which was maintained for 1 min; the temperature was then increased to 250 °C at a rate of 4 °C min^−1^ and held for 1 min; finally, the temperature was increased to 280 °C at a rate of 15 °C min^−1^ and held for 2 min.

Desorption was performed in spitless mode using a PSSI injector at 260 °C. A fused silica 30m × 0.25 mm × 0.25 μm film thickness SLB-5MS column was used for the analysis, and helium was used as a carrier gas at a constant flow of 1 mL min^−1^. Chromatograms were analyzed using AMDIS (National Institute of Standards and Technology, Gaithersburg, MD, United States), and compound identification was carried out by comparing detected spectra with NIST v. 2 (National Institute of Standards and Technology, United States). Only compounds with 80%, or higher, levels of matching were considered. The retention index (RI) information was used to increase the identification efficiency. VOCs were also compared with molecules in the literature on tomato fruit [35,42]. In order to reduce variability due to the SPME fiber decay, the raw peak area was normalized on the sum of the areas for each specific chromatogram.

### 3.5. Statistical Analysis

Data were processed by means of two-way ANOVA, using Statgraphics and considering the selenium concentration and time of its administration as variables. The results were then analyzed with the least significant difference (LSD) test (*p* < 0.05). Statistically significant values were indicated with different letters. VOCs were analyzed by partial least squares discriminant analysis (PLSDA) using JMP^®^, v. 16 (SAS Institute Inc., Cary, NC, USA).

## 4. Conclusions

Although less effective than supplementation via nutrient solution, selenium treatments performed by spraying tomato plants with 1 and 1.5 mg L^−1^ sodium selenate are effective in enriching tomatoes with Se concentrations, with no risks of toxicity for the human diet. In addition to the Se dosage, the developmental stage of the plant also seems to affect the biofortification effectiveness of the treatment with limited effects on fruit composition. Our approach to studying the physiological effect of Se-biofortification of fleshy fruit supports the evidence that the uptake of selenium causes metabolic changes in ripening fruit. This, then, results in increases in specific VOCs putatively associated with higher consumer preference scores. These changes in the aroma profile induced by Se represent a new and interesting aspect to study in future research, considering that the poor aroma of the commercial tomato varieties is one of the major causes of consumer complaints.

## Figures and Tables

**Figure 1 plants-10-01050-f001:**
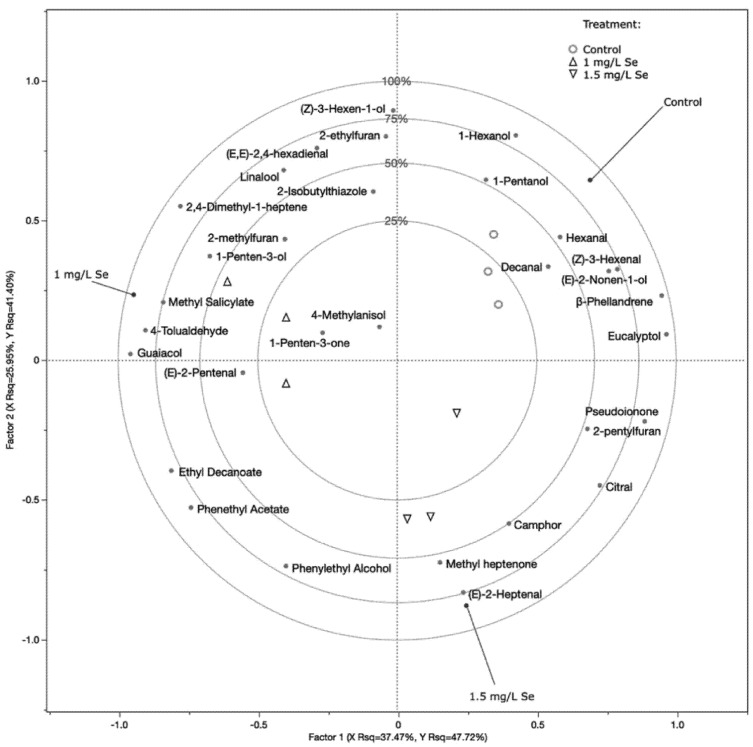
Partial least squares discriminant analysis (PLSDA). Treatment was employed as response variable, while the identified VOCs were used as predictor variables. Variable importance in projection (VIP) scores were used to filter compounds varying the most between different theses, and only compounds with a VIP score higher than 0.8 were included in the analysis. Circles, triangles, and inverted triangles were used to depict control fruit, 1 mg L^−1,^ and 1.5 mg L^−1^ sodium selenate-treated fruit, respectively.

**Table 1 plants-10-01050-t001:** Selenium concentration (µg kg^−1^ DW) in tomato fruit sprayed with sodium selenate (Na_2_SeO_4_) at different selenium concentrations and distributed at FL and fruit IG stages.

Treatment Time	Se Spray mg L^−1^	[Se] µg kg DW^−1^	
		1st Truss	2nd Truss
Flowering	0	0 a	0 a
	1.0	108 b	123 c
	1.5	146 b	131 c
Immature green	0	0 a	0 a
	1.0	92 b	84 b
	1.5	105 b	95.2 bc
Variance analysis			
Treatment time (A)		ns	***
Se dosage (B)		***	***
Interaction A × B		ns	***

Significance is as follows: ns, not significant; *** significant at 0.1%. Different letters in each column correspond to significantly different values for *p* < 0.05 according to the LSD (least significant difference) test.

**Table 2 plants-10-01050-t002:** Macro- and microelements concentration (mg kg^–1^) in tomato fruit of the first truss treated with sodium selenate (Na_2_SeO_4_) at different concentrations of Se at FL and fruit IG stages.

Treatment Time	Se Spray	Cu	Zn	Mn	Fe	K	Ca	Mg
	mg L^−1^	mg kg ^−1^						
Flowering	0	14.4 a	60.4 a	79.7 b	484 a	1672 a	429 ab	978 a
	1.0	13.5 a	65.5 a	88.6 b	493 a	1076 a	511 a	1007 a
	1.5	7.4 a	35.5 c	51.3 c	363 a	1540 ab	375 b	781 b
Immature green	0	14.1 a	60.2 a	80.0 b	479 a	1668 a	431 ab	980 a
	1.0	9.9 a	50.0 b	77.1 b	437 a	1374 b	458 a	909 a
	1.5	13.4 a	61.6 a	106 a	401 a	1005 c	540 a	1053 a
Variance analysis								
Treatment time (A)		ns	***	**	ns	ns	ns	**
Se dosage (B)		**	***	ns	ns	***	**	ns
Interaction A × B		**	***	***	ns	**	***	***

Significance is as follows: ns, not significant; ** significant at 1%; *** significant at 0.1%. Different letters in each column correspond to significantly different values for *p* < 0.05 according to the LSD (least significant difference) test.

**Table 3 plants-10-01050-t003:** Macro and microelements concentration (mg kg^–1^) in tomato fruit of the second truss treated with sodium selenate (Na_2_SeO_4_) at different concentrations of Se at FL and fruit IG stages.

Treatment Time	Se Spray	Cu	Zn	Mn	Fe	K	Ca	Mg
	mg L^−1^	mg kg ^−1^						
Flowering	0	14.3 a	67.1 a	95.0 a	550 a	1320 a	628 a	1074 a
	1.0	15.4 a	67.0 a	94.3 a	580 a	402 a	594 a	1054 a
	1.5	9.8 b	44.3 c	68.9 a	320 b	1355 a	516 b	851 b
Immature green phase	0	14.5 a	67.4 a	94.9 a	556 a	1327 a	630 a	1072 a
	1.0	8.9 b	39.8 c	55.2 a	392 b	1469 a	462 b	785 b
	1.5	10.5 b	53.8 c	84.1 a	366 b	1508 a	603 a	908 b
Variance analysis								
Treatment time (A)		***	***	ns	ns	ns	ns	ns
Se dosage (B)		*	***	**	**	ns	***	***
Interaction A × B		**	***	**	*	ns	***	**

Significance is as follows: ns, not significant; * significant at 5%; ** significant at 1%; *** significant at 0.1%. Different letters in each column correspond to significantly different values for *p* < 0.05 according to the LSD (least significant difference) test.

**Table 4 plants-10-01050-t004:** Incidence of blossom-end rot on fruits of plants treated with sodium selenate (Na_2_SeO_4_), at different selenium concentrations and distributed at FL and fruit IG stages.

Treatment Time	Se Spray mg L^−1^	Blossom−End Rot Incidence Affected/Total Fruit Ratio	
		1st Truss	2nd Truss
Flowering	0	25.3 a	40.4 a
	1.0	19.4 ab	33.0 ab
	1.5	14.4 ab	31.2 bc
Immature green	0	18.2 ab	36.8 a
	1.0	9.2 b	31.4 b
	1.5	9.6 b	20.2 c
Variance analysis			
Treatment time (A)		*	ns
Se dosage (B)		ns	*
Interaction A × B		ns	ns

Significance is as follows: ns, not significant; * significant at 5%. Different letters in each column correspond to significantly different values for *p* < 0.05 according to the LSD (least significant difference) test.

**Table 5 plants-10-01050-t005:** Qualitative characteristics of tomato fruits of the first truss treated with sodium selenate (Na_2_SeO_4_) at different concentrations of selenium and distributed at FL and fruit IG stages.

Treatment Time	Se Spray mg L^−1^	DW %	SSC °Brix	Titrable Acidity g Citric Acid 100 mL^−1^	Maturity Index	Taste Index
	0	5.0 c	6.5 a	0.8 a	8.0 a	1.1 a
Flowering	1.0	7.1 a	6.1 a	0.7 b	8.5 a	1 b
	1.5	6.5 a	6.4 a	0.7 b	9.3 a	0.9 b
	0	5.1 c	6.3 a	0.9 a	8.1 a	1 a
Immature green	1.0	6.7 a	6.4 a	0.8 ab	8.4 a	1 ab
	1.5	5.9 b	6.2 a	0.7 b	8.9 a	0.9 b
Variance Analysis						
Treatment time (A)		***	ns	ns	ns	ns
Se dosage (B)		***	ns	***	ns	**
Interaction A × B		*	ns	ns	ns	ns

Significance is as follows: ns, not significant; *, significant at 5%; **, significant at 1%; ***, significant at 0.1%. Different letters in each column correspond to significantly different values for *p* < 0.05 according to the LSD (least significant difference) test.

**Table 6 plants-10-01050-t006:** Qualitative characteristics of tomato fruits of the second truss treated with sodium selenate (Na_2_SeO_4_) at different concentrations of selenium and distributed at FL and fruit IG stages.

Treatment Time	Se Spray mg L^−1^	DW %	SSC °Brix	Titrable Acidity g Citric Acid 100 mL^−1^	Maturity Index	Taste Index
	0	4.5 bc	6.2 c	0.8 a	7.5 b	1 a
Flowering	1.0	5.4 a	7.0 a	0.7 a	9.4 a	1 a
	1.5	4.8 b	6.3 bc	0.6 b	10.2 a	0.8 c
	0	4.6 bc	6.0 c	0.8 a	7.6 b	1 a
Immature green	1.0	4.2 c	6.3 bc	0.7 a	8.8 a	0.9 b
	1.5	5.3 a	6.5 b	0.7 a	9.4 a	0.9 b
Variance Analysis						
Treatment time (A)		***	ns	ns	ns	ns
Se dosage (B)		***	**	**	*	**
Interaction A × B		***	***	***	ns	***

Significance is as follows: ns, not significant; *, significant at 5%; **, significant at 1%; ***, significant at 0.1%. Different letters in each column correspond to significantly different values for *p* < 0.05 according to the LSD (least significant difference) test.

## Data Availability

The data presented in this study are available on request from thecorresponding author.

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
