# Peer review of "Se-Enrichment Pattern, Composition, and Aroma Profile of Ripe Tomatoes after Sodium Selenate Foliar Spraying Performed at Different Plant Developmental Stages"

_plants, 2021, doi:10.3390/plants10061050_

Round 1
Reviewer 1 Report
The subject covered is of great practical interest. The authors treat it with great rigor, using appropriate tools and reach conclusions that from my point of view deserve to be published.
The introduction is clear, easy to read, with sufficient background and clear objectives.
Material and methods:
The methodology is correct, I only have a small doubt, which I am sure the authors will be able to clarify.
Why Material and Methods is point 3 and not point 2, which is the most frequent?
The results are clearly displayed.
The conclusions are based on the results.
References:
Check the assigned numbering. for example, I do not find the reference in the manuscript [6]
Author Response
Dear Reviewer,
The authors would like to thank you for your helpful comments and suggestions. The manuscript has been corrected in accordance with the comments:
The subject covered is of great practical interest. The authors treat it with great rigor, using appropriate tools and reach conclusions that from my point of view deserve to be published.
The introduction is clear, easy to read, with sufficient background and clear objectives.
Material and methods: The methodology is correct, I only have a small doubt, which I am sure the authors will be able to clarify.
Why Material and Methods is point 3 and not point 2, which is the most frequent?
- The manuscript has been formatted according to the Authors Guidelines of Plants by using the Microsoft Word template provided by the Journal.
The results are clearly displayed. The conclusions are based on the results.
References:
Check the assigned numbering. for example, I do not find the reference in the manuscript [6]
- The reference 6 has been added in the text (line 36), the numbering was corrected.
Reviewer 2 Report
There are only several minor comments to the manuscripts:
- Taking into account the novelty of the presented data it seems highly desirable to change the Title of the work indicating the effect of Se on tomato volatiles. That will increase the attractiveness of the work to the readers and indicate the significance of the work
The same for blouson-end incidence the 1st and the 2d trusses and different staged of biofortificatiobn
- Check references in the text: Line 32 change '4' to ‘[4]’.
The same at line 36 ‘[6]’ should be written
and line 38 ‘[8]’
line 48 ‘[14]’
line 177 ‘[41]’
- Line 39 ‘Those living in areas with a lack of selenium in the soil’- change to ‘with low selenium ‘
- Please decipher abbreviations the first time they appear in the text. For instance:
Line 83: what is FL and IG??
Line 148 SSC
Line 186 ‘PLSDA model’
- Line 110 ‘400 mcg’ has been already cited, see line 36
- Add units to Tables 4 and 5 (Titratable acidity, dry matter, incidence of blossom-end rot, SSC, Se)
- Line 104 ‘The selenium content in 100 g of fresh tomatoes was calculated by multiplying the concentration of Se by the percentage of dry matter. Values ranged between 0.62 and 0.95 µg Se 100 g FW-1 in the fruit of the first truss, and between 0.35 and 0.66 µg Se 100 g FW-1 in the fruit of the second truss. Thus, the daily consumption of treated tomatoes appears to increase the selenium supplementation, with no risk of toxicity for ’ – The sentance should be changed as the presented concentrations are too low even for RDA requirement.
- Line 235 and further: change mS/cm to mS cm-1
The same: lines 237-240: 4 mM/L N -NO3, 1 mM/L N-NH4, 1 mM/L P, 8 mM/L K, 5 mM/L Ca, 1.6 mM/L Mg, 5.23 mM/L Na, 1.71 mM/L S-SO4, 3.43 mM/L Cl, 15 μM/L Fe, 20 μM/L B, 1 mM/L Cu, 5 μM/L Zn, 10 μM/L Mn, and 1 μM/L Mo’ - change to 4 mM L-1; etc
and lines 291-292, 295: ‘4 °C/min… 15 °C/min…. 1 mL/min’ change to 4 oC min-1… 15 oC min-1… 1 mL min-1)
- Make changes in the reference list according to the authors rules (for instance: ‘Journal Articles:
Author 1, A.B.; Author 2, C.D. Title of the article. Abbreviated Journal Name Year, Volume, page range.’; etc)
Author Response
Dear Reviewer,
The authors would like to thank you for your helpful comments and suggestions. The manuscript has been corrected in accordance with the comments:
Taking into account the novelty of the presented data it seems highly desirable to change the Title of the work indicating the effect of Se on tomato volatiles. That will increase the attractiveness of the work to the readers and indicate the significance of the work. The same for blouson-end incidence the 1st and the 2d trusses and different staged of biofortification.
- We thank the Referee for the useful suggestion. The title has been changed accordingly.
Check references in the text: Line 32 change '4' to ‘[4]’.
The same at line 36 ‘[6]’ should be written
and line 38 ‘[8]’
line 48 ‘[14]’
line 177 ‘[41]’
- All the references numbers in the text are now between brackets.
Line 39 ‘Those living in areas with a lack of selenium in the soil’- change to ‘with low selenium ‘
- The text has been changed.Please decipher abbreviations the first time they appear in the text. For instance:
Line 83: what is FL and IG??
Line 148 SSC
Line 186 ‘PLSDA model’
- All the abbreviations have been defined in the text.
Line 110 ‘400 mcg’ has been already cited, see line 36
- We agree with the Referee and we deleted the sentence.
Add units to Tables 4 and 5 (Titratable acidity, dry matter, incidence of blossom-end rot, SSC, Se)
- Units have been added.
Line 104 ‘The selenium content in 100 g of fresh tomatoes was calculated by multiplying the concentration of Se by the percentage of dry matter. Values ranged between 0.62 and 0.95 µg Se 100 g FW-1 in the fruit of the first truss, and between 0.35 and 0.66 µg Se 100 g FW-1 in the fruit of the second truss. Thus, the daily consumption of treated tomatoes appears to increase the selenium supplementation, with no risk of toxicity for ’ – The sentance should be changed as the presented concentrations are too low even for RDA requirement.
- We agree with the referee that the Se-enriched tomatoes in our trial do not satisfy the RDA requirements for selenium. We wanted just to point out that the consumption of these Se-enriched tomatoes appears to be beneficial for increasing the selenium supplementation.
Line 235 and further: change mS/cm to mS cm-1
The same: lines 237-240: 4 mM/L N -NO3, 1 mM/L N-NH4, 1 mM/L P, 8 mM/L K, 5 mM/L Ca, 1.6 mM/L Mg, 5.23 mM/L Na, 1.71 mM/L S-SO4, 3.43 mM/L Cl, 15 μM/L Fe, 20 μM/L B, 1 mM/L Cu, 5 μM/L Zn, 10 μM/L Mn, and 1 μM/L Mo’ - change to 4 mM L-1; etc
and lines 291-292, 295: ‘4 °C/min… 15 °C/min…. 1 mL/min’ change to 4 oC min-1… 15 oC min-1… 1 mL min-1)
- The text has been corrected.
Make changes in the reference list according to the authors rules (for instance: ‘Journal Articles:
Author 1, A.B.; Author 2, C.D. Title of the article. Abbreviated Journal Name Year, Volume, page range.’; etc)
- References have been corrected.